# BALMS: Benchmarking Agentic LLMs for Mental Health Sensing

Yu Yvonne Wu [1]   Arvind Pillai [1]   Yuliang Chen [1]   Sudarshan Regmi [1]   Tess Z. Griffin [1]   Michael V. Heinz [1]
Lisa Marsch [1]   Nicholas C. Jacobson [1]   Andrew Campbell [1]

## Abstract

Ubiquitous wearable and mobile devices can collect large-scale, longitudinal multivariate behavioral and physiological signals, opening new opportunities for personalized monitoring of mental wellbeing. LLM-based agentic systems, which augment language models with reasoning, planning, and external tools, offer a promising paradigm for interpreting such longitudinal records and providing personalized, interactive health support. However, most existing healthcare agents are designed for clinical text and have not been systematically evaluated on daily-life multivariate sensor data, leaving the capabilities and limitations of agentic paradigms on this structured longitudinal modality unclear. We present a benchmark of LLM-based agentic systems for longitudinal wearable mental wellbeing tasks, spanning multiple datasets, using both closed and open source backbones, and three representative agentic frameworks. Through extensive evaluation on mental health prediction tasks and sensitivity analyses over temporal length, we jointly report performance, cost, and latency, and distill design insights for agentic systems on longitudinal mental health sensing and support.

## 1. Introduction

Mental wellbeing, encompassing symptoms of depression, anxiety, and stress, is a growing public health concern. Traditional assessment relies on episodic, in-clinic instruments that place a high burden on patients and providers (Substance Abuse and Mental Health Services Administration, 2025). Wearable and mobile devices offer a promising alternative, passively and continuously collecting longitudinal, multivariate behavioral and physiological signals that capture day-to-day variation that clinic visits miss (Xu et al.,

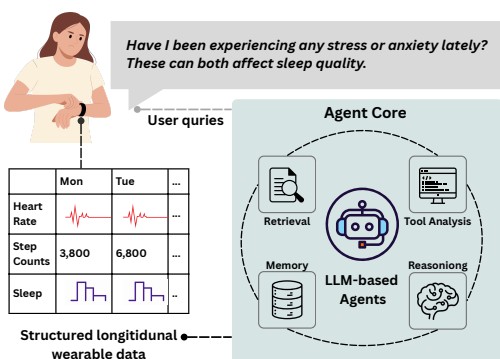

*Figure 1.* An LLM-based agentic system for longitudinal mental health sensing.

2023; Nepal et al., 2024; Gomes et al., 2023). This opens a path toward proactive, personalized mental health monitoring and support (Xu et al., 2023; Nepal et al., 2024; Gomes et al., 2023). However, realizing this potential requires the ability to interpret long, multi-channel and irregular signals within the context of an individual's daily life and self-reports.

One promising avenue for addressing this interpretation challenge is to augment large language models (LLMs) with reasoning, planning, and tool use, resulting in agentic systems. Recent healthcare agent benchmarks show that such systems can flexibly handle a wide range of clinical and biomedical tasks, including diagnostic question answering, clinical reasoning, evidence retrieval, and structured record interpretation, and that they generalize across heterogeneous data schemas with little task-specific tuning (Schmidgall et al., 2025; Arora et al., 2025). By coupling pretrained reasoning with retrieval and memory, agents can also produce personalized, context-aware outputs such as explanations, recommendations, and conversational follow-ups grounded in a user's own history, which fits the multi-cohort, multi-instrument reality of mobile sensing (Kim et al., 2026).

Despite these advancements, agentic systems for longitudinal mobile mental health remain underexplored. Existing efforts fall broadly into two categories: (i) clinical-text agents that operate over structured medical records or clinical notes (Schmidgall et al., 2025), and (ii) a small but growing set of mobile health LLM applications that answer short, single-window queries over fixed wearable devices (Choube et al., 2025; Heydari et al., 2025). However, neither line

[1]Dartmouth College, Hanover, NH, USA. Correspondence to: Yu Yvonne Wu <yvonne.wu@dartmouth.edu>.

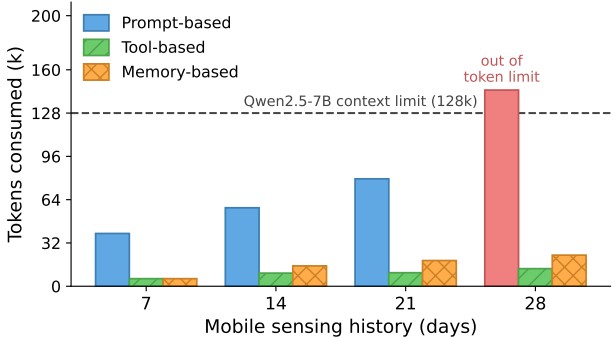

*Figure 2.* Token consumption of 3 agent paradigms as the mobile sensing history grows from 7 to 28 days on stress detection task.

of work targets longitudinal multivariate sensing regimes, which introduces distinct challenges. First, ingesting days to weeks of multi-channel, high-sampling-rate sensor data quickly exhausts the LLM context window (Figure 2). Second, numeric reasoning over long time-series arrays is empirically unreliable (Pillai et al., 2025; Spathis & Kawsar, 2024). Finally, inference cost and latency scale unfavorably with the observation horizon, and no standardized protocol exists for comparing agent designs along these axes.

These observations motivate the first comprehensive benchmark of LLM-based agentic systems for longitudinal mobile health, with a focus on daily mental wellbeing prediction tasks. Building on existing mobile health LLM systems, we evaluate three representative paradigms, prompt-based, tool-based, and memory-based, on three real-world longitudinal datasets (*DiversityOne, PMData, GLOBEM*) with multiple sensor modalities, and observation horizons (from months to years), benchmarking on both open-source (*Qwen*) and closed (*Claude*) backbones. Our analysis demonstrates how each paradigm trades off accuracy and efficiency, and shows that as look-back window grows, prompt-based ingestion degrades, retrieval-based memory improves, and tool-based access stays flat, pointing to memory-based designs as a promising path for longitudinal grounding.

## 2. Benchmarking LLM Agents for Mental Health Sensing

In longitudinal mental health sensing, a user carries a smartphone and a wearable device that passively and continuously collect behavioral and physiological data over weeks, months, or even years. Multiple times a day, the user is prompted to complete an ecological momentary assessment (EMA), a self-reported survey that captures their current mental state in their natural environment and serves as the gold-standard label for mental health research (Shiffman et al., 2008). For instance, depression symptoms are measured through the Patient Health Questionnaire (PHQ-9) (Kroenke et al., 2001), while anxiety is assessed using the Generalized Anxiety Disorder scale (GAD-7) (Spitzer

et al., 2006). As this setting is common across mental health sensing studies, we formulate our problem as follows: given an input window of multivariate sensing time-series data, predict the user's mental health condition label. For example, given step count, heart rate, and calorie data, predict the user's level of depression, anxiety, or stress. Toward understanding the capabilities of LLM-based agentic systems for such prediction tasks, we benchmark three families of methods that have driven recent progress: (i) prompt-based, (ii) tool-augmented, and (iii) memory-augmented methods.

### 2.1. Prompt-based Reasoning Systems

To predict mental health symptoms, existing LLM approaches start with prompt-based reasoning, directly probing the pretrained knowledge of the LLM. The input numerical data is transformed into a text prompt, and the model is asked to produce a prediction (Kim et al., 2024). These approaches demonstrate that LLMs can perform inference on sensing data to produce predictions and explanations through carefully designed prompts and chain-of-thought reasoning (Gruver et al., 2023). **Health-LLM** (Kim et al., 2024) is the most comprehensive baseline in this family and is originally evaluated across multiple wearable datasets and a range of mental-wellbeing prediction targets. It formats each sensor modality as a per-day numeric array spanning the look-back window, includes the user's demographic profile and a task-specific instruction (*e.g.*, the score range and the question being asked) as additional context, and assembles them into a single prompt that queries the LLM to output the target label as an integer. We adopt this prompt template directly and only vary the look-back window and the underlying backbone; the full template is provided in Appendix A. Such prompt-only systems consume a fixed context window and do not retrieve memory, call tools, or plan analyses, which motivates the stronger agentic paradigms.

### 2.2. Tool-based Agents

Tool-based methods extend the agentic system with executable tools such as code interpreters (Allan et al., 2010). These tools, including rolling statistics, trend extraction, and modality-wise summaries, let LLMs compute over long, multi-channel sensing data rather than directly read them. Recent passive-sensing tool-based methods (PHIA (Merrill et al., 2026), GLOSS (Choube et al., 2025), PHA (Heydari et al., 2025)) follow this template with different tool sets, but are typically tied to a specific sensor schema and have not been benchmarked across different cohorts. **PHIA** (Merrill et al., 2026) is a representative tool-based system for passive-sensing analysis that generalizes across different sensor setups. It preloads each user's record as a set of in-memory pandas Dataframes and drives the LLM agents through a ReAct loop (Yao et al., 2022): at each step, the model emits a [Thought] → [Act] → Python code trajectory,

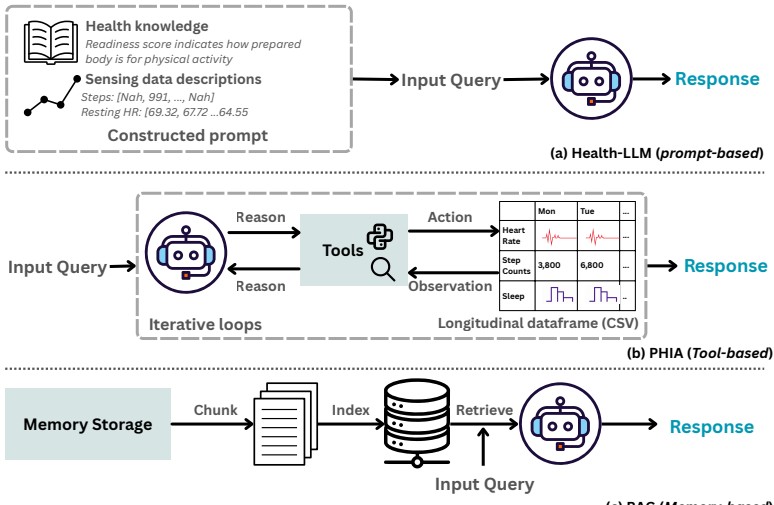

*Figure 3.* Overview of the three agentic system benchmarked: prompt-based, tool-based, and memory-based.

where the generated code consists of pandas operations such as date filtering, groupby aggregation, executed against the preloaded Dataframes. The resulting numeric output is appended to the trajectory and fed back into the next step until the agents output a final answer (Appendix A).

### 2.3. Memory- and Retrieval-augmented Agents

A complementary way to handle long histories is to keep the past in an external memory store and retrieve only the relevant entries into the prompt at inference time (Wang et al., 2023). The agent then reasons only over the entries of the user's history most relevant to the current query, selected by content similarity rather than recency. To our knowledge, no memory-augmented agent (Chhikara et al., 2025) has been developed for mental-health sensing; we therefore start from a standard retrieval-augmented generation (RAG) recipe and adapt it to the wearable-sensing setting. **Day-level RAG.** Standard text RAG chunks documents at the token or paragraph level (Jin et al., 2025), which does not align with the granularity of mobile sensing, where both the self-reported labels and the sensor aggregates are naturally daily. We therefore chunk memory at the day level rather than at the text-token level. Concretely, each historical day's sensing data are encoded with a sentence-transformer (Reimers & Gurevych, 2019) into a memory bank. At inference, we retrieve the top-$k$ similar days by cosine similarity and pass only those days, together with the query to the LLM.

## 3. Experiments and Results

**Datasets and Tasks.** We evaluate 3 agentic systems on 3 publicly available longitudinal passive-sensing datasets, summarized in Table 1. **Base LLM selection.** We evaluate three backbones: two open-source (*Qwen2.5-7B-Instruct* and *Qwen2.5-14B-Instruct*) and one closed-source (*Claude-*

*Table 1.* **Datasets statistics.**

| Datasets | DiversityOne (Busso et al., 2025) | PMData (Thambawita et al., 2020) | GLOBEM (Xu et al., 2023) |
|---|---|---|---|
| Avg Length | 28 days | 5 months | multi-year |
| # Sensors | 10 | 5 | 8 |
| Participants | 782 | 16 | 497 |
| Sensors | Accelerometer, Gyroscope, Wi-Fi, etc | Steps, Resting HR, Burned Calories, etc | Step, Sleep, Accelerometer, etc |
| Tasks | Emotion prediction | Stress detection | Anxiety detection |

*Haiku-4.5*) (Anthropic, 2025). **Metrics.** We use MAE (mean absolute error) for mental health scores, and also report latency (s) representing running time of each sample. For each system we follow the official implementation guidelines (Appendix A).

### 3.1. Zero-shot mental health detection

This task tests whether each agentic system can predict daily mental-health labels (Emotion, Stress, Anxiety) from longitudinal multimodal signals in a zero-shot regression setting, with no per-task fine-tuning. Table 2 reveals three patterns. **(i) Backbone scaling helps prompt- and memory-based agents but not tool-based ones.** Health-LLM and RAG improve consistently from Qwen2.5-7B to Qwen2.5-14B to Claude-Haiku on every dataset, whereas PHIA improves only on PMData (0.64→0.54) and degrades on DiversityOne (1.22→2.11 at 14B) and GLOBEM (1.41→1.52). For tool-based agents the bottleneck is program reliability rather than textual reasoning, and stronger LLMs tend to write more elaborate code that compounds errors on unfamiliar schemas. **(ii) PHIA's gains generalize only when the data schema matches its training distribution.** PMData's Fitbit-style daily aggregates mirror the dataframe schema PHIA's ReAct pipeline was originally tuned on, so the LLM-generated pandas code targets familiar columns; on DiversityOne's raw mobile streams (accelerometer, gyroscope, Wi-Fi) and GLOBEM's sparser step/sleep records, the same pipeline generates schema-sensitive code that amplifies errors and does not benefit from a stronger backbone.

*Table 2.* **Zero-shot mental health detection.** We compare three agentic systems with common open-source and closed-source backbones. Lower is better for MAE, Latency, and Tokens (per sample). The best result per dataset is in **bold** and the second-best is underlined.

| Backbone | Method | DiversityOne(Mood) | | | PMData(Stress) | | | GLOBEM(Anxiety) | | |
|---|---|---|---|---|---|---|---|---|---|---|
| | | MAE$^\downarrow$ | Latency (s)$^\downarrow$ | Token$^\downarrow$ | MAE$^\downarrow$ | Latency (s)$^\downarrow$ | Token$^\downarrow$ | MAE$^\downarrow$ | Latency (s)$^\downarrow$ | Token$^\downarrow$ |
| *Open-source LLMs* | | | | | | | | | | |
| Qwen2.5-7B | Health-LLM | 0.60 | 9.2 | 30530 | 0.65 | 7.5 | 31572 | 1.17 | 1.5 | 5455 |
| | RAG | 0.96 | 4.5 | 15403 | 0.72 | 2.6 | 8611 | 1.20 | 1.6 | 1094 |
| | PHIA | 1.22 | 11.1 | 19788 | 0.64 | 30.0 | 17035 | 1.48 | 28.2 | 7810 |
| Qwen2.5-14B | Health-LLM | 0.44 | 16.7 | 30530 | 0.62 | 10.4 | 31572 | 0.86 | 2.2 | 5455 |
| | RAG | 0.53 | 6.6 | 15403 | 0.71 | 4.1 | 8611 | 0.90 | 1.7 | 1094 |
| | PHIA | 2.11 | 18.9 | 19788 | 0.57 | 17.6 | 17035 | 1.52 | 32.4 | 7810 |
| *Closed LLMs* | | | | | | | | | | |
| Claude-Haiku-4.5 | Health-LLM | **0.41** | 8.9 | 30530 | 0.64 | 4.3 | 31572 | **0.83** | 1.8 | 5400 |
| | RAG | 0.87 | 5.4 | 15403 | 0.67 | 2.5 | 8470 | 0.96 | 1.2 | 1250 |
| | PHIA | 1.11 | 19.6 | 19670 | **0.54** | 18.6 | 16435 | 1.41 | 13 | 12181 |

As a result, PHIA's per-dataset ranking flips: best on PM-Data (MAE 0.54) but worst on DiversityOne (1.11–2.11) and GLOBEM (1.41–1.52). **(iii) Efficiency separates the paradigms more cleanly than accuracy.** RAG is consistently the cheapest (∼1–15k tokens, 1.2–5.4s) because only the top-$k$ retrieved days reach the LLM; Health-LLM's prompt holds the full history and grows with sensor count and horizon (∼30k tokens on DiversityOne and PMData); PHIA's prompt is moderate but its multi-call ReAct loop dominates wall time (up to 32s, ∼10× RAG). The same accuracy is therefore reachable at very different operating points: PHIA trades latency for accuracy, while RAG matches Health-LLM at a fraction of the tokens.

### 3.2. Longitudinal length discussion

We test how each paradigm reasons over increasingly long histories on PMData with Qwen2.5-14B. We fix the target day and vary the look-back window from 2 weeks to the full 5 months. The three paradigms exhibit qualitatively different scaling performance (Figure 4). **Health-LLM** is most accurate at the shortest window (0.39 at 2 weeks) but worsens steadily as more history is appended; we hypothesize that the textualised history crowds out the target day's cues and stresses numeric reasoning over long arrays (Spathis & Kawsar, 2024). **RAG** shows the opposite trend, improving from 0.74 to 0.53 as a memory bank encompasses longer sensing data, turning more history directly into better grounding. **PHIA** is essentially flat (0.41–0.44) across all windows. It queries a preloaded dataframe through generated code, so the look-back length does not change the per-step input. However, this robustness comes at a cost that token use and latency stay high at every horizon (Figure 5). Overall, long-horizon reasoning is paradigm-dependent: longer prompts degrade performance, retrieval accuracy benefits from richer memory, and tool-based ac-

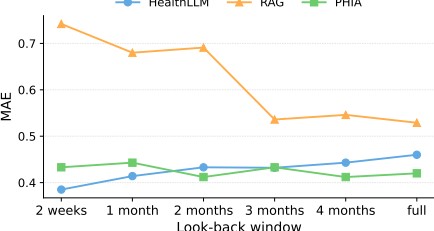

*Figure 4.* Effect of look-back window length on PMData.

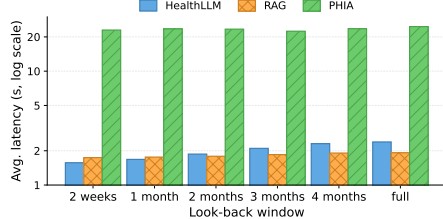

*Figure 5.* Efficiency analysis on various look-back window length.

cess remains stable with high latency. This suggests longitudinal grounding requires structured history, not simply longer context.

## 4. Conclusion

This work evaluates three representative agentic-LLM paradigms on longitudinal mobile sensing for mental health, an axis that prior agent benchmarks have not exercised. Across PMData, GLOBEM, and DiversityOne we find that the three paradigms scale very differently with look-back length. Specifically, prompts saturate, retrieval improves, and tool-based access stays flat, suggesting that effective grounding requires structured access to the past rather than longer prompts. We will extend this study with reasoning-trace and open-ended qualitative analyses, and explore memory-based agent designs that turn long passive histories into reliable health-state grounding.

## Acknowledgements

The authors acknowledge support for this research from Evergreen: A Generative AI & Behavioral Sensing Digital Ecosystem to Promote Student Wellness and Flourishing. This work is made possible through philanthropic gifts to Dartmouth College dedicated to advancing AI-supported well-being and flourishing of college students.

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

# A. Appendix

---

**Health-LLM Prompt Template (Stress Prediction)**

*Predict the user's stress level given wearable-derived physiological signals and self-reported context.*

**Task instruction.**
You are a personalized healthcare agent trained to predict {Stress} which ranges from {0} to {5}.

**User profile.**
The user is a {48} years old {male} with {195} height with a measured maximum heart rate of {182} bpm.

**Sensor signals (look-back window).**
**[Steps]**: [4664, 3035, NaN, 1284, 4966, NaN, NaN, 2094, 4185, 4698, 3384, 867, NaN, 7742] steps
**[Burned Calories]**: [494, 354, NaN, 145, 517, NaN, NaN, 252, 454, 470, 339, 104, NaN, 1120] calories
**[Resting Heart Rate]**: [53.7411, 52.8815, 53.222, 54.3111, 52.2591, 54.2032, 53.09, 53.3421, 53.0407, 52.4206, 52.5101, 52.1151, 51.8893, 51.9333] beats/min
**[SleepMinutes]**: [NaN, 430, 821, NaN, 362, 1096, 392, NaN, 391, 422, 779, NaN, 361, 367] minutes
**[Mood]**: 3 out of 5

**Question.**
What would be the predicted stress?

**Output format.**
For example, the answer should be in the following format:
Answer: 1

---

**PHIA Template (Stress Prediction)**

**Example ReAct trace.**
```
[step 1: model]
```
**[Thought]**: I need to analyze the 14 days of prior sensor data to predict the PHQ-4 anxiety sub-scale score. Let me start by exploring the structure of the data and computing relevant features that might correlate with anxiety.
**[Act]**: tool_code
```
# First, let's examine the structure and date range of the data
print("Data shape:", subday_df.shape)
print("\nFirst few rows:")
print(subday_df.head())
```
*[+243 chars]*

```
[step 1: observe]
Data shape: (56, 10)
First few rows:
date      segment   steps   sleep_minutes   sleep_efficiency   screen_unlock ...
0 2018-05-02  afternoon   NaN            NaN                NaN              88.025 ...
1 2018-05-02  evening     NaN            NaN                NaN              60.770 ...
```
*[+1644 chars]*

```
[step 2: model]
```
**[Thought]**: Good, I have 14 days of data (May 2–15) with 4 segments per day. Most columns have many NaN values. Let me compute daily aggregates and look for patterns that might correlate with anxiety, such as sleep quality, screen time, mobility, and social activity.
**[Act]**: tool_code
```
# Aggregate to daily level
daily_df = subday_df.groupby('date').agg({
'steps': 'sum',
'sleep_minutes': 'sum', ...})
```
*[+391 chars]*

```
[step 2: observe]
Code executed (no output).
```
*(Loop continues for up to M iterations until a final* Answer: *is emitted.)*

---

**Datasets and tasks.** *DiversityOne* (Busso et al., 2025) is a multimodal mobile sensing dataset collected from 782 participants over a 28-day window, capturing phone-derived behavioral signals such as accelerometer, gyroscope, and Wi-Fi traces alongside daily emotion self-reports. *PMData* (Thambawita et al., 2020) combines five months of Fitbit-derived physiological signals (steps, resting heart rate, burned calories) from 16 participants with daily lifestyle and wellness self-reports, originally introduced for sports and lifestyle logging. *GLOBEM* (Xu et al., 2023) is a multi-year longitudinal benchmark from 497 college students that pairs continuous step and sleep tracking with regular mental-health self-reports. Two of these (PMData and GLOBEM) are also used in the original Health-LLM benchmark (Kim et al., 2024), providing continuity with the prompt-based literature, while DiversityOne extends the evaluation to a larger and more demographically diverse cohort.

**Tasks.** Each dataset contributes one zero-shot prediction task targeting a distinct facet of mental wellbeing: *Emotion* (DiversityOne), *Stress* (PMData), and *Anxiety* (GLOBEM). Following Health-LLM (Kim et al., 2024), every task is framed as a regression over an integer-valued self-report scale; given the user's longitudinal signals over a look-back window, the agentic system must emit a single integer prediction.

**Implementation details.** All baselines are run following the official guidelines. For the prompt-based system, we adopt the Health-LLM template verbatim. For the tool-based system (PHIA), the ReAct loop is capped at 5 iterations per query, and the in-memory pandas dataframe is initialized with the user's full longitudinal record. For the memory-based system, we retrieve the top-3 most similar days using LangChain standard framework and concatenate them with the query day before passing the prompt to the inference LLM. Open-source models are served via vLLM on $4\times$ A6000 GPUs.

