# OpenReview forum: "BALMS: Benchmarking Agentic LLMs for Mental Health Sensing"
_ICML.cc/2026/Workshop/FMSD — FMSD @ ICML 2026 Poster_

### Official Review · Reviewer_eAxP · 2026-05-21
**BALMS: Benchmarking Agentic LLMs for Mental Health Sensing**

**Rating:** 8
**Confidence:** 4

**Review:**

## Summary

BALMS benchmarks three agentic LLM paradigms, prompt-based (Health-LLM), tool-based (PHIA), and memory-based (RAG) ,on longitudinal mental-health prediction from passive wearable sensing, across three datasets and three backbones. The headline finding is paradigm-dependent scaling with look-back window: prompts degrade, retrieval improves, tools stay flat.

## Strengths

- Very good problem setup and clear articulation of the gap around non-text, longitudinal sensor data analysis.
- Clearly written and comprehensively experimented paper.

## Areas for Improvement

- No non-LLM baseline, which makes the absolute MAE numbers in Table 2 uninterpretable.
- Figure 4: unclear why increasing context hurts Health-LLM and PHIA, sometimes monotonically. The paper hypothesizes prompt crowding for Health-LLM but does not explain why PHIA (which queries a preloaded dataframe) degrades at all.
- Figure 5: PHIA's latency appears flat across look-back windows and Health-LLM/RAG barely move either, which is surprising since longer histories should mean larger prompts, more ReAct iterations, or larger memory banks.